# Early Maladaptive Schemas and Self-Stigma in People with Physical Disabilities: The Role of Self-Compassion and Psychological Flexibility

**DOI:** 10.3390/ijerph191710854

**Published:** 2022-08-31

**Authors:** Anna Pyszkowska, Monika M. Stojek

**Affiliations:** 1Department of Social Sciences, University of Silesia in Katowice, 40-007 Katowice, Poland; 2Department of Psychiatry and Behavioral Sciences, Emory University School of Medicine, Atlanta, GA 30322, USA

**Keywords:** disability, early maladaptive schemas, self-stigma, psychological flexibility, self-compassion

## Abstract

Self-stigmatizing thoughts may be rooted in one’s core beliefs, and in turn be associated with early maladaptive schemas (EMS). Psychological flexibility, an ability to distance and accept one’s thoughts, is reported to diminish EMS’s effect on well-being, while self-compassion, a mindful attitude towards one’s suffering, often reduces self-stigma. The objective of this study was to examine associations between EMS, self-stigma, psychological flexibility and self-compassion in individuals with disabilities, as they are at higher risk of experiencing self-stigma. Participants were 238 persons with disabilities. The Self-Stigma Scale, Young’s Schemas Questionnaire, the Self-Compassion Scale Short and the Acceptance and Action-II Questionnaire were used. Hierarchical regression and mediation analysis were used to establish (1) predictors and (2) potential mediators of self-stigma in people with disabilities. Hierarchical regression showed that EMS alone accounted for 39% of the variance explained by self-stigma, and with the addition of psychological flexibility—an additional 2% was explained. Parallel mediation analyses indicated that psychological flexibility partially mediated the relationship between EMS domains and self-stigma. It appears that psychological rigidity is related to self-stigma and should be addressed in treatment through evidence-based approaches such as Schema Therapy and Acceptance and Commitment Therapy to enhance individuals’ healthy life patterns, flexibility and self-compassion.

## 1. Introduction

There are two crucial contexts of the experience of disability. The first is related to disability per se, understood as the lack of or a significant decrease in physical, sensory, or mental abilities resulting in health, social and/or work dissatisfaction that impairs normal functioning [1]. The popularity of the medical model of the disease has increased the importance of the second context, which is the social perception of people with disabilities. Ableism refers primarily to the belief that disability as such is undesirable, unfavorable or destructive, and, if possible, should be treated, repaired or even eliminated [2]. It is also related to a set of beliefs and behaviors that favor unequal treatment of people because of their apparent disability, resulting in stigmatization. Research has shown that social exclusion and social perception are important elements of the experience of disability and that these systemic and social aspects often play a role far more crucial in terms of the well-being of an individual than the disability itself [3]. Stigmatized individuals anticipate rejection and discrimination due to stigma, which, in turn, leads them to increased social isolation and decreased employment, leisure and social opportunities, and these perceived “failures” may result in decreased self-efficacy, self-esteem and self-worth [4]. External stigma is then transferred into internal, affective-cognitive processes, resulting in internal stigma, referred to as “self-stigma” [5].

Watson and Larson [4] indicated that the awareness of stigma is not synonymous with internalizing it as people who belong to a stigmatized group are aware of stereotypes but may not agree with them. Thus, self-stigma appears only when an individual agrees with stigmatizing opinions and applies them to the self (e.g., “people with disabilities are dependent on others, weak or unattractive” = “I am dependent on others, weak or unattractive as a person with a disability”) as the nature of self-stigma is defined as rigid, narrow [6], self-referring and self-directed [7]. Additionally, the experience of stigma and self-stigma is heterogeneous as it takes various forms depending on the nature of the condition and specific aspects of a certain disability or illness [8,9]. According to the Corrigan and Watson model [10], (perceived, enacted) public stigma and (felt, internalized) self-stigma are two separate constructs, which exhibit only moderate associations. Research shows that self-stigma acts as a more destructive stressor compared to public stigma (conceptualized as “minority stress”) [11].

Self-stigma is reported to demonstrate a wide spectrum of effects on affective, cognitive and behavioral functioning of individuals in various stigmatized groups. Previous studies on mental health stigma and self-stigma showed associations with shame and self-criticism [8,12], low self-esteem [13], diminished life satisfaction and a sense of meaning in life [6,7,8]. A significantly lower number of studies on self-stigma among people with chronic illness and disabilities also demonstrated links with low self-esteem and self-efficacy, negative cognitions about oneself, e.g., “I am not capable”, “I am a weak person” [14], shame [15], diminished quality of life [16,17] and tendencies towards social isolation [15,18,19]. Kowalski and Peipert [20] obtained interesting results in terms of differentiation between people with physical and psychological disabilities and their experience of stigma and self-stigma. Although the first group gained higher results in perceived public stigma, both groups reported similar self-stigma rates. In all reports, self-stigma acts as a thinking pattern that affected various domains of individuals’ cognitive schemas related to dependence, weakness and social inadequacy [6,21].

Early maladaptive schemas (EMS) are defined as “extremely stable and enduring themes, comprised of memories, emotions, cognitions and bodily sensations regarding oneself and one’s relationship with others that develop during childhood and are elaborated on throughout the individual’s lifetime, and that are dysfunctional to a significant degree” [22] and act as a fundamental construct in Jeffrey Young’s schema therapy (ST). Young et al. [22] identified eighteen schemas, grouped into four domains [23]. The characteristics of EMS are summarized in Table 1.

Young et al. [22] highlighted the links between EMS and displaying maladaptive coping strategies. It can be emphasized that EMS may be interpreted as core beliefs [24] or cognitive fusion [25] as one automatically filters thoughts, emotions and behaviors through the lenses of a particular schema, resulting in maladaptive reactions and distress. As there is no literature on linking EMS and self-stigma, it can be hypothesized that the latter is linked to schemas regarding Disconnection and Impaired Autonomy. Research depicts self-stigma as a depressive attributional style resulting in thoughts concentrated on being ashamed of stigma, feelings of self-worthlessness, inferiority and weakness [26,27], similar to Defectiveness/Shame schema, resulting in tendencies towards dissociation [28,29] (Emotional Inhibition schema), feelings of lack of autonomy and being dependent on others [30] (Dependence schema), social withdrawal, exclusion and isolation [29,31] (Social Isolation schema) and lack of support and understanding [29,30] (Emotional Deprivation and Abandonment/Instability schemas).

Wood, Byrne and Morrison [32] stressed that self-stigma was associated with negative core beliefs related to early aversive experiences. Although research on protective factors connected with defectiveness-related EMS is scarce, a growing body of research emphasizes the role of self-compassion and psychological flexibility in decreasing the impact of self-stigma and negative self-concept. Self-compassion is defined as a gentle attitude towards oneself in times of suffering and acceptance of one’s own difficulties whilst they are considered a common human experience [33]. Psychological flexibility is described as the ability to stay in contact with the present moment regardless of unpleasant thoughts, feelings and sensations, while choosing and developing one’s behavior repertoire based on personal values and situational contexts [34]. Self-compassion is reported to favor coping and the quality of life among people with mental [35,36] and chronic illness [37,38] and disabilities [39], as well as to act as a protective factor against self-stigma [40,41,42]. Psychological flexibility also shows negative associations with self-stigma [12,36], positive associations with adaptive emotional schemas [43] and acts as a resilience factor in individuals with chronic conditions [44]. Additionally, self-stigma is related to experiential avoidance, one of the processes of psychological rigidity [45], resulting in passivity, obtaining avoidant-style coping strategies [46] and diminished adherence to psychiatric and psychotherapeutic treatment [47]. Chan et al. [36] proposed a mindfulness model of stigma resistance that demonstrated psychological flexibility and self-compassion as protective factors against self-stigma and facilitators of life satisfaction in a psychiatric sample.

While direct positive relationships between psychological flexibility and self-compassion have been widely established [48,49,50], limited research showed negative associations between EMS and psychological flexibility [51] and mindful attention awareness [52,53,54]. As mindfulness is a component of the self-compassion model [33], it can be hypothesized that similar relationships would be established between EMS and self-compassion itself. Fischer et al. [51] demonstrated that EMS increased the extent of cognitive fusion and experiential avoidance. It is hypothesized that psychological rigidity is rooted in early maladaptive schemas [35].

For the purpose of the current study, two objectives were indicated. The first and main aim was to identify the relationships between EMS and self-stigma among people with disabilities. Despite growing interest in the area of self- and affiliate-stigma, there has been no research on cognitive and emotional schemas and self-stigmatizing beliefs in this sample. On the basis of the literature cited, it can be hypothesized that self-stigma would be associated with two EMS domains, i.e., Disconnection and Impaired Autonomy, as the experience of disability stigmatization is often associated with beliefs concentrated on shame, social isolation and dependence [6,22].

The second aim of this study was to verify whether self-compassion and psychological flexibility may act as potential mediators between EMS and self-stigma. According to several reports [22,40,41,43,55], self-compassion and psychological flexibility acted as important buffers against self-stigma. They were also negatively associated with EMS [38,39]. It was hypothesized that self-compassion and psychological flexibility would act as at least partial mediators between schemas from Disconnection and Impaired Autonomy domains and self-stigma, decreasing the impact of EMS on self-stigmatizing beliefs.

## 2. Materials & Methods

### 2.1. Participants

Nonprobability sampling was used. The participants were recruited in Poland via the Internet on various forums and groups for patients and people with disabilities, using a snowball sampling. The inclusion criterion was diagnosed disability (physical, sensory, genetic or other). All participants provided written informed consent prior to enrolment in the study. The study was voluntary.

Two hundred and thirty-eight people with disabilities participated in the study (68% female, M_age_ = 37.89 years, SD_age_ = 17.04 years) of whom 63.87% were subjects with acquired disability; 35.52% declared physical disability, 18.99% sensory disability, 15.19% neurological disease (including genetic disorders), 10.97% metabolic disease, 5.91% cancer, 5.48% cardiovascular disease and 7.94% reported “another disease or disability”.

### 2.2. Measurements

To measure the variables of interest, socio-demographic metrics and four questionnaires (YSQ-S3-PL, Self-Stigma Scale, AAQ-II, Self-Compassion Scale Short) were used. Participants who decided to be involved in the study self-completed the following tools:

Early maladaptive schemas. The Polish adaptation of the Young Schema Questionnaire-Short Form 3 (YSQ-S3-PL [56]) by Oettingen, Chodkiewicz, Mącik and Gruszczyńska [57] was used. It consists of 90 items (e.g., ‘People have not been there to meet my emotional needs’) rated on a Likert-type scale from 1 (completely untrue of me) to 6 (describes me perfectly). It measures the following eighteen EMS: Emotional deprivation (α = 0.81), Abandonment/Instability (α = 0.82); Mistrust/Abuse (α = 0.83); Social Isolation (α = 0.86); Defectiveness/Shame (α = 0.88); Failure (α = 0.86); Dependence/Incompetence (α = 0.71); Vulnerability To Harm Or Illness (α = 0.79); Enmeshment/Undeveloped Self (α = 0.82); Subjugation (α = 0.76); Self-sacrifice (α = 0.72); Emotional Inhibition (α = 0.81); Unrelenting Standards (α = 0.69); Entitlement/Grandiosity (α = 0.62); Insufficient Self-Control/Self-Discipline (α = 0.78); Approval-Seeking/Recognition-Seeking (α = 0.82); Pessimism (α = 0.82) and Punitiveness (α = 0.79). Questionnaire overall Cronbach’s α in the present sample was α = 0.97. Descriptions of scales and the domains they form are described in Table 1.

Self-stigma. The Self-Stigma Scale by Mak and Cheung [58] was used. It was adapted into Polish for the purpose of the study. The scale consists of nine items (e.g., ‘My identity as a person with a disability is a burden to me’) rated on a Likert-type scale from 1 (strongly disagree) to 4 (strongly agree). Scale Cronbach’s α in the present study was α = 0.89.

Psychological flexibility. The Acceptance and Action Questionnaire (AAQ-II) by Bond et al. [59] was used. It was adapted into Polish by Kleszcz et al. [60]. Psychological flexibility is measured with the use of seven questions (e.g., ‘I’m not afraid of my feelings’) rated on a Likert-type scale from 1 (always untrue) to 7 (always true). Questionnaire Cronbach’s α in the present study was α = 0.89.

Self-compassion. The Self-Compassion Short Scale [61] was used. It was adapted into Polish by Kocur et al. [62]. The scale consists of 12 items (e.g., ‘I try to see my failings as part of the human condition’) rated on a scale from 1 (almost never) to 5 (almost always). Scale Cronbach’s α in the present sample was α = 0.7.

### 2.3. Statistical Analysis

Participants with acquired and innate disabilities were compared on demographic variables using the Mann–Whitney U test and on variables of interest using the univariate ANOVA. Correlations were conducted using the Spearman rank correlation analysis. To examine relative contributions of early maladaptive schemas, cognitive flexibility and self-compassion to the prediction of self-stigma, two hierarchical linear regressions were conducted. In the first regression, 18 maladaptive schemas were entered in the first step, followed by cognitive flexibility in the second, and self-compassion in the third step. In the second regression, four domains of early maladaptive schemas were entered in the first step, followed by cognitive flexibility in the second, and self-compassion in the third step. Multicollinearity was examined using the tolerance and variance inflation factor (VIF).

To test whether the relationship between early maladaptive schemas and self-stigma is mediated by cognitive flexibility and self-compassion, a bootstrapped parallel mediation analysis with 95% confidence interval (CI) with 5000 resamples was conducted [63]. In a parallel multiple mediator model, the predictor variable X is assumed to directly influence the outcome variable Y as well as indirectly through two or more mediators (M1, M2), and the mediators are assumed to not causally influence one another. For these analyses, we reported standardized estimates as well as unstandardized coefficients, *B*, to indicate the relative strengths of mediation relationships (also called indirect effects). To reduce the number of tests conducted, only early maladaptive schema domains were tested as predictor variables in the model, and tests were limited to those domains which demonstrated significant relationships with self-stigma in a hierarchical linear regression model.

Calculations were made using the JASP 0.12.2.0 statistical package (University of Amsterdam, Amsterdam, The Netherlands, 2018) and SPSS PROCESS macro version 4 (IBM, Amunk NY USA 2020) [63]. For all statistical tests, an α level of 0.05 was considered to be statistically significant.

## 3. Results

The Mann–Whitney U test showed no significant differences (*p* >0.05) between people with acquired and innate disabilities. Additionally, ANOVA tests showed no differentiation (*p* > 0.05) regarding various types of disabilities in terms of the variables studied. Therefore, in subsequent analyses, the type or character of disability was not differentiated in the results.

Table 2 presents descriptive statistics and the Spearman rank correlation analysis.

The obtained results indicated significant relationships (*p* < 0.001) between self-stigma and each EMS. The highest coefficients were obtained between self-stigma and schemas from Impaired Autonomy (Dependence/Incompetence, *r* = 0.52; Failure To Achieve, *r* = 0.43; Vulnerability To Harm Or Illness, *r* = 0.36) and Disconnection (Defectiveness/Shame, *r* = 0.45; Social Isolation, *r* = 0.39; Emotional Deprivation, *r* = 0.38) domains, while Impaired Limits (Entitlement/Grandiosity, *r* = 0.13, Approval-Seeking, *r* = 0.15) showed the lowest relationships with self-stigma.

Each schema was characterized by showing a medium or high negative correlation with both psychological flexibility (range from *r* = −0.26 to *r* = −0.64) and self-compassion (range from *r* = −0.13 to *r* = −0.58). Schemas related to self-victimization or self-stigmatization demonstrated the highest negative relationships with both resources: Defectiveness/Shame (for P-F *r* = −0.62, for S-C *r* = −0.58), Failure to Achieve (for P-F *r* = −0.64, for S-C *r* = −0.57), Abandonment/Instability (for P-F *r* = −0.62, for S-C *r* = −0.55), Vulnerability To Harm Or Illness (for P-F *r* = −0.57, for S-C *r* = 0.49) and Subjugation (for P-F *r* = −0.63, for S-C *r* = −0.56). Additionally, self-stigma showed medium negative relationships with psychological flexibility (*r* = −0.43) and self-compassion (*r* = −0.36).

A summary of the hierarchical linear regression analyses is shown in Table 3 and Table 4. The collinearity diagnostics indicated satisfactory tolerance and VIF indicators (i.e., tolerance > 0.2 and VIF < 5) [64].

The results revealed that eighteen early maladaptive schemas accounted for 39% of the variance explained (F = 7.768, *p* < 0.001) by self-stigma. Two schemas acted as significant predictors, i.e., Social Isolation (β = 0.20, *p* < 0.05) and Dependence (β = 0.41, *p* < 0.05). After controlling for the main effects of EMS, psychological flexibility and self-compassion, EMS and psychological flexibility (β = −0.22, *p* < 0.05) explained an additional 2% of the variance in self-stigma (R^2^ = 0.42, *p* < 0.001). The addition of self-compassion did not have any effect on the variance. However, while using four domains of EMS as potential predictors of self-stigma, it was revealed that only two acted as significant predictors: Disconnection (β = 0.21, *p* < 0.001) and Impaired Autonomy (β = 0.28, *p* < 0.001), and the model accounted for 26% of variance explained (F = 22.175, *p* < 0.001). In this model, additional predictors in the form of psychological flexibility and self-compassion did not act as significant predictors of self-stigma. The results obtained in both models may be considered as consistent as Social Isolation and Dependence are facets of Disconnection and Impaired Autonomy.

Based on the results obtained in the regression model, two mediation analyses were performed: one including the EMS domain of Impaired Autonomy as the predictor variable, and the other including the Disconnection domain. The schemata of the tested parallel mediation model are presented in Figure 1. The results of the analyses are summarized in Table 5.

The analyses revealed that the Disconnection domain was significantly positively associated with self-stigma (c = 3.02, CI 2.29, 3.74), even when controlling for the mediators (c’ = 1.85, CI 0.84, 2.84), consistent with a partial mediation. Similarly, the Impaired Autonomy domain was significantly positively associated with self-stigma (c = 3.58, CI 2.80, 4.36), even when controlling for the mediators (c’ = 2.59, CI 1.48, 3.71), consistent with a partial mediation. In both cases, only psychological flexibility acted as a significant mediator. Given that psychological flexibility was negatively related to Disconnection (*B* = −6.97, CI −7.87, −6.07) and to self-stigma (*B* = −0.14, CI −0.26, −0.03), the indirect effect analysis of Disconnection on self-stigma via psychological flexibility (ab = 1.00, CI 0.12, 1.95) indicated that individuals who are one unit higher on the Disconnection domain are also estimated to be one unit higher on self-stigma, as a result of the effect of higher Disconnection on lower psychological flexibility, which in turn affects higher self-stigma. Similarly, the indirect analysis of Impaired Autonomy on self-stigma via psychological flexibility (ab = 0.96, CI 0.09, 1.87) indicated that individuals who are one unit higher on the Impaired Autonomy domain, are estimated to be 0.96 units higher on self-stigma, as a result of the effect of higher Impaired Autonomy on lower psychological flexibility, which in turn affects higher self-stigma.

## 4. Discussion

People with disabilities are often stereotyped as the ones with diminished autonomy and independence [30], and disability itself may be recognized as an undesirable and unfavorable shameful malfunction [2]. The obtained results are in line with these assumptions as EMS of the highest relationships with self-stigma represented the Impaired Autonomy and Disconnection domains. It can be assumed that these results are associated with the roots of self-stigma in negative core beliefs about oneself [32] and public stigma [4] which functions as a model for negative and diminishing opinions towards a stigmatized trait or an individual. High associations with Defectiveness/Shame (*r* = 45), Dependence (*r* = 53) and Failure to Achieve (*r* = 43) schemas exemplified a self-referencing and self-directing nature of self-stigma and the relationships with Social Isolation (*r* = 39) and Abandonment (*r* = 35) schemas highlighted the role of social and public spheres in applying the stigma to the self. As a result, it can be assumed that EMS related to self-stigma covered both sides of its nature, i.e., external and internal, which is in accordance with previous research depicting self-stigma as a result of public stigma applied to oneself [4] and self-directedness [7].

It was demonstrated that self-stigma showed negative relationships with self-compassion (*r* = −0.36) and psychological flexibility (*r* = −0.43), which is in accordance with several previous research reports [21,40,41,55]. It can be hypothesized that self-stigmatizing opinions and beliefs are the exemplification of automatic thinking patterns [32] and the main function of self-compassion and psychological flexibility is to loosen the rigidity of the content of these patterns, as well as to increase the ability of distancing from negative self-concept and widening the spectrum of self-perception beyond disability or another stigmatizing social role. Additionally, both psychological flexibility and self-compassion exhibited moderate or high negative relationships with EMS (especially with the Disconnection and Impaired Limits domains), which is in accordance with previous research [51,52,53].

Hierarchical linear regression indicated that all eighteen EMS predicted 39% of self-stigma (with Social Isolation and Dependence as significant predictors), and psychological flexibility added 2%, thus increasing the overall variance explained to 41% (self-compassion remained as an insignificant predictor). Of note, EMS consist of eighteen variables that did not constitute the entire variance when psychological flexibility was added to the model, which may suggest that the latter is a variable robust on its own, playing an imperative role in predicting self-stigma. This result complements the current knowledge in this field as the direct relationships between these variables have not been proven so far. It may be hypothesized that the link between self-stigma and psychological rigidity (or lack of flexibility) could be due to strong fusion to one’s self-stigmatizing assumptions about oneself. Cognitive fusion, understood as the tendency for behavior to be overly regulated and influenced by cognition, may be exemplified by treating one’s thoughts about oneself as a rule that one has to follow [65]. For example, fusion with self-stigmatizing thoughts may lead to actions and behaviors consistent with stigmatizing belief, e.g., fusion with “People will not like me because of my disability” may result in self-isolation.

The parallel mediation analyses indicated that psychological flexibility (but not self-compassion) partially mediated the relationship between the Disconnection and Impaired Autonomy EMS domains, and self-stigma. The highest mediating effect was obtained in the Impaired Autonomy schema domain, and it can be hypothesized that the component of committed action of psychological flexibility, promoting activity and living in accordance with one’s values, can play a role in decreasing self-stigma related to experiential avoidance [45]. On the other hand, partial mediation between Disconnection and self-stigma could be related to self-as-context psychological flexibility that encourages a non-judgmental and non-labeling way of perceiving oneself as a complex, multilayered being that cannot be defined by one or two traits (in that case, “disability” would be an insufficient description of one’s identity). Psychological flexibility played a dominant role in both regression and mediation analyses, encompassing self-compassion in terms of both predicting self-stigma and acting as a mediator between certain EMS domains and self-stigma. These results are in line with reports by Davey et al. [49] who demonstrated that self-compassion was no longer significant when psychological flexibility-related variables were added to their model regarding pain interference and adaptation in patients experiencing chronic pain. Furthermore, Harris [65] stressed that self-compassion can be considered, in fact, the seventh domain of the hexaflex (the six-part model of psychological flexibility), as self-kindness, mindfulness and common humanity, which are three dimensions of self-compassion, are an integral part of psychological flexibility.

Despite strengths, such as an adequate sample size (*N* = 238), the present study has limitations. First, the study design was cross-sectional, hence, it must be highlighted that longitudinal studies concerned with the development of self-compassion and psychological flexibility are required in order to provide an evaluation of causal relationships among the variables studied. While the mediational model assumes causality between variables and is optimally tested using longitudinal or experimental designs, Hayes [63] points out that one can conduct a mediational analysis even when one cannot unequivocally establish causal relationships between variables, given a solid plausible theoretical foundation. Therefore, despite the fact that data in the current study were collected at one time point, a strong case is made for the sequential order of the variables of interest. Second, the current study depicts a widely heterogeneous group of people with various disabilities. Therefore, it is vital to perform further research on narrower and specific samples in order to verify whether self-stigma and EMS vary among different groups in terms of the type or visibility of a given disability (in the current study, the ANOVA test showed no significant differences; however, it might have been due to the unequal sample distribution). This provides opportunities for future research on more specific groups (perhaps also in the control group consisting of people without disability).

## 5. Conclusions

The current study was designed to verify associations between self-stigma and early maladaptive schemas and the role of psychological flexibility and self-compassion in a group of people with physical disabilities. The results obtained revealed that self-stigma is associated with early maladaptive schemas and can be interpreted as an example of psychological rigidity and negative automatic thoughts. Therefore, the main function of a compassionate attitude towards oneself and psychological flexibility may be to loosen the rigidity of self-stigmatizing contents. These findings should be relevant for clinical research and practice as both Schema Therapy [22] and Acceptance and Commitment Therapy [34] are evidence-based therapeutic approaches that aim to work on individuals’ thoughts and enhancing one’s healthy life patterns, flexibility and self-compassion.

Of note, the current study is the first to show associations among unhelpful schemas, psychological flexibility and self-stigma in a population at risk for self-stigma—individuals with disabilities. Due to the fact that EMS are the representation of core, cognitive-oriented beliefs, it would be recommended to widen the scope of research and study the relationships between self-stigma and emotional schemas. Additionally, the current study provides an applicable basis for interventions for people with disabilities struggling with self-stigma as psychological flexibility is proved to act as a buffer against self-stigma and negative automatic thoughts.

## Figures and Tables

**Figure 1 ijerph-19-10854-f001:**
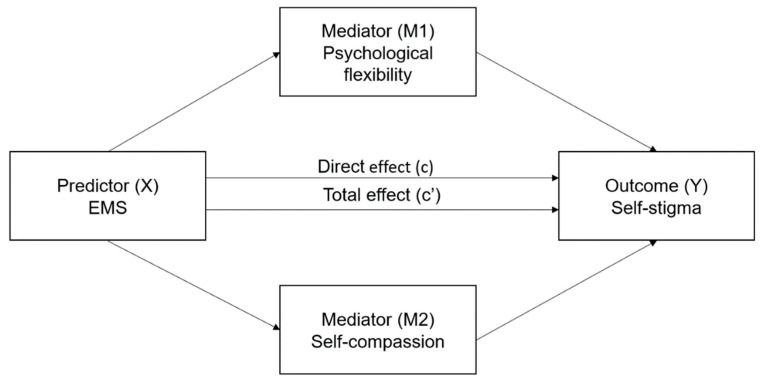
The parallel mediation model.

**Table 1 ijerph-19-10854-t001:** Characteristics of early maladaptive schemas (EMS), see: Young et al., 2003 [22].

Domain/Schema	Characteristics
**Disconnection**	
Emotional Deprivation	Expectation that one’s desire for a normal degree of emotional support will not be adequately met by others.
Emotional Inhibition	The excessive inhibition of spontaneous action, feeling or communication—usually to avoid disapproval by others, feelings of shame or losing control of one’s impulses.
Mistrust/Abuse	The expectation that others will hurt, abuse, humiliate, manipulate or take advantage.
Social Isolation	The feeling that one is isolated from the rest of the world, different from other people and/or not part of any group or community.
Defectiveness/Shame	The feeling that one is defective, unlovable, bad, unwanted, inferior or invalid in important respects.
**Impaired Autonomy**	
Subjugation	Excessive surrendering of control to others because one feels coerced—usually to avoid anger, retaliation or abandonment.
Dependence	Belief that one is unable to handle one’s everyday responsibilities in a competent manner, without considerable help from others (e.g., take care of oneself, solve daily problems, exercise good judgment, tackle new tasks, make good decisions).
Failure to Achieve	The belief that one has failed, will inevitably fail or is fundamentally inadequate relative to one’s peers, in areas of achievement (school, career, sports, etc.).
Vulnerability To Harm Or Illness	Exaggerated fear that imminent catastrophe (e.g., medical, emotional, external) will strike at any time and that one will be unable to prevent it.
Abandonment/Instability	The perceived instability or unreliability of those available for support and connection.
Enmeshment	Excessive emotional involvement and closeness with one or more significant others (often parents), at the expense of full individuation or normal social development.
Insufficient Self-control	Pervasive difficulty or refusal to exercise sufficient self-control and frustration tolerance to achieve one’s personal goals, or to restrain the excessive expression of one’s emotions and impulses.
**Impaired Limits**	
Entitlement	The belief that one is superior to other people, entitled to special rights and privileges, or not bound by the rules of reciprocity that guide normal social interaction.
Approval-Seeking	Excessive emphasis on gaining approval, recognition or attention from other people, or fitting in, at the expense of developing a secure and true sense of self.
**Exaggerated Standards**	
Self-Sacrifice	Excessive focus on voluntarily meeting the needs of others in daily situations, at the expense of one’s own gratification.
Unrelenting Standards	The underlying belief that one must strive to meet very high internalized standards of behavior and performance, usually to avoid criticism. Typically results in feelings of pressure or difficulty slowing down and in hypercriticalness toward oneself and others.
Pessimism	A pervasive, lifelong focus on the negative aspects of life (pain, death, loss, disappointment, conflict, guilt, resentment, unsolved problems, potential mistakes, betrayal, things that could go wrong, etc.) while minimizing or neglecting the positive or optimistic aspects.
Punitiveness	The belief that people should be harshly punished for making mistakes.

**Table 2 ijerph-19-10854-t002:** Descriptive Statistics and Spearman’s rank correlation coefficient, * *p* < 0.001.

	M	SD	Self-Stigma	Psychological Flexibility	Self-Compassion
Disconnection	2.46	1.05	0.43 *	−0.70 *	−0.57 *
Emotional Deprivation	2.16	1.15	0.38 *	−0.52 *	−0.39 *
Emotional Inhibition	2.68	1.25	0.29 *	−0.58 *	−0.50 *
Mistrust/Abuse	2.67	1.22	0.30 *	−0.60 *	−0.46 *
Social Isolation	2.67	1.36	0.39 *	−0.56 *	−0.50 *
Defectiveness/Shame	2.11	1.23	0.45 *	−0.62 *	−0.58 *
Impaired Autonomy	2.62	0.92	0.45 *	−0.71 *	−0.64
Subjugation	2.35	1.07	0.34 *	−0.63 *	−0.56 *
Dependence/Incompetence	2.28	1.02	0.52 *	−0.47 *	−0.42 *
Failure To Achieve	2.52	1.24	0.43 *	−0.64 *	−0.57 *
Vulnerability To Harm Or Illness	2.82	1.27	0.36 *	−0.57 *	−0.49 *
Abandonment/Instability	3.08	1.30	0.35 *	−0.62 *	−0.55 *
Enmeshment	2.19	1.18	0.33 *	−0.40 *	−0.34 *
Insufficient Self-Control	3.10	1.18	0.21 *	−0.47 *	−0.42 *
Impaired limits	3.21	0.78	0.06	−0.11	0.10
Entitlement/Grandiosity	3.07	0.96	0.13 *	−0.26 *	−0.20 *
Approval-Seeking	3.34	1.21	0.15 *	−0.44 *	−0.41 *
Exaggerated standards	3.20	0.87	0.39 *	−0.58 *	−0.51 *
Self-Sacrifice	3.52	1.04	0.22 *	−0.25 *	−0.13 *
Unrelenting Standards	3.39	1.07	0.29 *	−0.42 *	−0.40 *
Pessimism	3.19	1.28	0.34 *	−0.63 *	−0.53 *
Punitiveness	2.72	1.10	0.36 *	−0.46 *	−0.46 *
Self-stigma	18.13	6.68	-	−0.43 *	−0.36 *
Psychological flexibility	31.03	10.35	−0.43 *	-	0.69 *

**Table 3 ijerph-19-10854-t003:** Hierarchical regression analyses, * *p* < 0.05, *** *p* < 0.001.

Step	Predictor	Unstandardized Coefficients		Standardized Coefficients	R^2^	R^2^ Change	F
		B	SE	β			
1.					0.39	0.39	7.768 ***
	Emotional Deprivation	0.61	0.48	0.11			
	Emotional Inhibition	−0.24	0.44	−0.04			
	Mistrust/Abuse	−0.77	0.55	−0.14			
	Social Isolation	1.00	0.47	0.20 *			
	Defectiveness/Shame	0.89	0.53	0.16			
	Subjugation	−0.12	0.60	−0.18			
	Dependence/Incompetence	2.69	0.58	0.41 *			
	Failure To Achieve	−0.30	0.52	−0.06			
	Vulnerability To Harm Or Illness	0.58	0.47	0.11			
	Abandonment/Instability	0.06	0.44	0.01			
	Enmeshment	0.03	0.41	0.01			
	Insufficient Self-Control	0.09	0.44	0.01			
	Entitlement/Grandiosity	−0.06	0.52	−0.01			
	Approval-Seeking	−0.44	0.46	−0.08			
	Self-Sacrifice	0.31	0.42	0.05			
	Unrelenting Standards	0.76	0.51	0.12			
	Pessimism	0.29	0.55	0.06			
	Punitiveness	0.07	0.50	0.01			
2.					0.41	0.02	7.962 ***
	Emotional Deprivation	0.58	0.47	0.10			
	Emotional Inhibition	−0.38	0.43	−0.07			
	Mistrust/Abuse	−0.86	0.54	−0.16			
	Social Isolation	0.99	0.46	0.20 *			
	Defectiveness/Shame	0.71	0.53	0.13			
	Subjugation	−1.15	0.59	−0.18 *			
	Dependence/Incompetence	2.78	0.57	0.42 *			
	Failure To Achieve	−0.62	0.53	−0.12			
	Vulnerability To Harm Or Illness	0.50	0.46	0.09			
	Abandonment/Instability	−0.21	0.45	−0.04			
	Enmeshment	0.18	0.41	0.03			
	Insufficient Self-Control	−0.04	0.44	−0.01			
	Entitlement/Grandiosity	0.07	0.51	0.01			
	Approval-Seeking	−0.50	0.45	−0.09			
	Self-Sacrifice	0.33	0.41	0.05			
	Unrelenting Standards	0.59	0.50	0.09			
	Pessimism	0.14	0.54	0.03			
	Punitiveness	0.21	0.49	0.03			
	Psychological flexibility	−0.15	0.06	−0.23 *			
3.					0.41	0.00	7.537 ***
	Emotional Deprivation	0.60	0.47	0.10			
	Emotional Inhibition	−0.38	0.43	−0.07			
	Mistrust/Abuse	−0.85	0.55	−0.15			
	Social Isolation	0.99	0.46	0.20 *			
	Defectiveness/Shame	0.69	0.54	0.12			
	Subjugation	−1.16	0.59	−0.18 *			
	Dependence/Incompetence	2.78	0.57	0.42 *			
	Failure To Achieve	−0.63	0.53	−0.12			
	Vulnerability To Harm Or Illness	0.49	0.46	0.09			
	Abandonment/Instability	−0.23	0.45	−0.04			
	Enmeshment	0.18	0.41	0.03			
	Insufficient Self-Control	−0.05	0.44	−0.01			
	Entitlement/Grandiosity	0.08	0.51	0.01			
	Approval-Seeking	−0.51	0.45	0.09			
	Self-Sacrifice	0.35	0.42	0.05			
	Unrelenting Standards	0.57	0.51	0.09			
	Pessimism	0.14	0.55	0.03			
	Punitiveness	0.21	0.49	0.03			
	Psychological flexibility	−0.14	0.06	−0.22 *			
	Self-compassion	−0.02	0.06	−0.02			

**Table 4 ijerph-19-10854-t004:** Hierarchical regression analyses, * *p* < 0.05, *** *p* < 0.001.

Step	Predictor	Unstandardized Coefficients		Standardized Coefficients	R^2^	R^2^ Change	F
		B	SE	β			
1.					0.26	0.26	22.175 ***
	Disconnection	0.35	0.57	0.21 ***			
	Impaired Autonomy	2.05	0.70	0.28 ***			
	Impaired Limits	−0.50	0.49	−0.06			
	Exaggerated Standards	0.73	0.64	0.09			
2.					0.28	0.02	18.395 ***
	Disconnection	1.01	0.60	0.16			
	Impaired Autonomy	1.66	0.74	0.23 *			
	Impaired Limits	−0.41	0.50	−0.05			
	Exaggerated Standards	0.63	0.64	0.08			
	Psychological flexibility	−0.09	0.05	−0.14			
3.					0.28	0.00	15.263 ***
	Disconnection	1.01	0.60	0.16			
	Impaired Autonomy	1.67	0.75	0.23 *			
	Impaired Limits	−0.41	0.50	−0.05			
	Exaggerated Standards	0.63	0.64	0.08			
	Psychological flexibility	−0.09	0.06	−0.14			
	Self-compassion	0.01	0.06	0.02			

**Table 5 ijerph-19-10854-t005:** Two parallel bootstrapped mediation models testing whether the relationship between either Disconnection domain (X) of early maladaptive schemas (EMS) or Impaired Autonomy domain (X) of EMS, and self-stigma (Y) is mediated by Psychological Flexibility (M1) and/or Self-compassion (M2).

Model & Effects	X: Disconnection				Impaired Autonomy			
	Standardized Estimate	Unstandardized Effect *B* (SE)	95% CI		StandardizedEstimate	Unstandardized Effect *B* (SE)	95% CI	
			Lower	Upper			Lower	Upper
Direct effect of X on Y	**0.29**	**1.85 (0.51)**	0.84	2.86	**0.37**	**2.59 (0.57)**	1.48	3.71
Total effect of X on Y	**0.47**	**3.02 (0.37)**	2.29	3.74	**0.51**	**3.58 (0.39)**	2.80	4.36
Indirect effects:								
Total indirect effect of X on Y	**0.18**	**1.17 (0.37)**	0.46	1.90	**0.14**	**0.99 (0.43)**	0.17	1.85
X->M1->Y	**0.16**	**1.00 (0.46)**	0.12	1.95	**0.14**	**0.96 (0.45)**	0.09	1.87
X->M2->Y	0.03	0.16 (0.30)	−0.45	0.74	0.01	0.03 (0.35)	−0.64	0.73

Effects at the statistically significant level (*p* ≤ 0.05) are bolded.

## Data Availability

Publicly available datasets were analyzed in this study. These data can be found here: https://osf.io/6g2wn/?view_only=ff23de187f514d3e99cac480540f223c (accessed on 10 August 2022).

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
