# Peer review of "Early Maladaptive Schemas and Self-Stigma in People with Physical Disabilities: The Role of Self-Compassion and Psychological Flexibility"

_ijerph, 2022, doi:10.3390/ijerph191710854_

Round 1

Reviewer 1 Report

The aim of the present study was to identify the relationship between early maladaptive schemas and self-stigma and further examine the mediating role played by self-compassion and psychological flexibility using a cross-sectional design. The sample of this online survey was comprised of people diagnosed with disability only (n = 238). Correlation, hierarchical regression and mediation analyses were performed for data analysis. results showed that among the eighteen maladaptive schemas, social isolation and dependence acted as significant predictors of self-stigma while among the four domains of maladaptive schemas, disconnection and impaired autonomy acted as significant predictors. Results of parallel mediation showed that for both domains of disconnection and impaired autonomy, only psychological flexibility partially mediated between EMS and self-stigma. The current study contributes to a better understanding of the relationship between EMS, psychological flexibility, self-compassion and self-stigma in theory. Also, this study has practical implications for developing interventions for people with disability in the future.

The authors present an important question concerning the self-stigma of people with disability and have drawn interesting conclusions. Despite this, I have some questions and suggestions on this study.

First, I think the writing logic from line 95 to line 120 is a little bit messed. I can see the authors want to clarify the potential mediation role of psychological flexibility and self-compassion in these two paragraphs but I feel this was not stated clearly. In the former paragraph, the authors (the dependent variable) and introduce the definition of self-compassion and psychological flexibility, but then the authors provide evidence to support the relationship between psychological flexibility, self-compassion and EMS (the independent variable) next. And in the latter paragraph, the authors demonstrate how psychological flexibility and self-compassion may influence self-stigma (the dependent variable). In order to make the logic smooth, I recommend that after suggesting the role of psychological flexibility and self-compassion in reducing self-stigma, the authors illustrate how this could happen and provide support for this argument. Then, the authors may explain how EMS is related to psychological flexibility and self-compassion and finally propose the hypothesis.

Second, from line 104 to line 107, evidence was provided to support the relationship between EMS and self-compassion, but actually, the authors state the negative association between EMS and mindful attention awareness in this article. Mindful attention awareness is not equal to self-compassion so I doubt the rationality of the evidence. The authors may explain that mindfulness is one component of self-compassion and this evidence could support the relationship between EMS and self-compassion to some extent.

Third, I notice that the Materials & Methods part does not include the introduction and description of participants and the authors leave it to the Result part. Though acceptable, it’s still recommended that introducing participants in the Materials & Methods part.

Fourth, I have a question about the process of hierarchical linear regression. In the current study, the authors establish the hypothesized model with two parallel mediations so I wonder why the authors enter psychological flexibility and self-compassion into the regression one by one rather than enter them both at one time.

Fifth, I recommend the authors check the writings of this article carefully before submitting it. A spelling error “inte4rval” is detected at line 179. Moreover, in Table 2, the correlation between self-stigma and defectiveness/shame is in an incorrect format and needs to be adjusted.

Author Response

We would like to thank the editor and the reviewers for their time and consideration. All comments and suggestions on the article helped us to make changes, which, we hope, will improve its content. Please find below a list of changes against each point which had been raised.

1.    The suggestion was addressed in the text as the paragraphs order was changed (lns 95-123): “Wood, Byrne and Morrison [32] stressed that self-stigma was associated with negative core beliefs related to early aversive experiences. Although research on protective factors connected with defectiveness-related EMS is scarce, a growing body of research emphasizes the role of self-compassion and psychological flexibility in decreasing the impact of self-stigma and negative self-concept. Self-compassion is defined as a gentle attitude towards oneself in times of suffering and acceptance of one’s own difficulties whilst they are considered a common human experience [33]. Psychological flexibility is described as the ability to stay in contact with the present moment regardless of unpleasant thoughts, feelings, and sensations, while choosing and developing one's behavior repertoire based on personal values and situational contexts [34]. Self-compassion is reported to favor coping and the quality of life among people with mental [43, 44] and chronic illness [44, 45] and disabilities [46], as well as to act as a protective factor against self-stigma [47-49]. Psychological flexibility also shows negative associations with self-stigma [43, 46, 12], positive with adaptive emotional schemas [50], and acts as a resilience factor in individuals with chronic conditions [51]. Additionally, self-stigma is related to experiential avoidance, one of the processes of psychological rigidity [52], resulting in passivity, obtaining avoidant-style coping strategies [53] and diminished adherence to psychiatric and psychotherapeutic treatment [54]. Chan et al. [43] proposed a mindfulness model of stigma resistance that demonstrated psychological flexibility and self-compassion as protective factors against self-stigma and facilitators of life satisfaction in a psychiatric sample.

While direct positive relationships between psychological flexibility and self-compassion have been widely established [35-37], limited research showed negative associations between EMS and psychological flexibility [38] and mindful attention awareness [39-41]. As mindfulness is a component of the self-compassion model [33], it can be hypothesized that similar relationships would be established between EMS and self-compassion itself. Fischer et al. [38] demonstrated that EMS increased the extent of cognitive fusion and experiential avoidance. It is hypothesized that psychological rigidity is rooted in early maladaptive schemas [42].

2.  An additional explanation was inserted in lines 107-109: “As mindfulness is a component of the self-compassion model [33], it can be hypothesized that similar relationships would be established between EMS and self-compassion itself”.

3. The section “2.1. Participants” was included under the Materials & Methods section.

4. In addition to the hypothesis regarding two parallel mediations, another goal of the study was to examine the contribution of psychological flexibility and self-compassion to self-stigma separately – adding them both at once would make it impossible to check their specific influence on the dependent variable.

5. The paper has been proofread for errors.

Reviewer 2 Report

Thank you for the opportunity to review the paper entitled “Early Maladaptive Schemas and Self-Stigma in People with Physical Disabilities. The Role of Self-Compassion and Psychological Flexibility”. The manuscript is well-written so I only have a few minor comments:

1.   The title used a full-stop after “Disabilities”. It would be better if the authors use “Disabilities: The Role of Self-Compassion and Psychological Flexibility”.

2.   Line 137: The authors indicated that four questionnaires were used. Could the authors please clarify whether they used four questionnaires or a questionnaire with four different measurements.

3.   Under the section of Materials and methods, the authors did not indicate how they recruited the participants and who the participants were. Instead, the authors reported some information about the participants in the results section. The authors can consider how to report the details of the recruitment of participants etc in the appropriate section.   

4.   Lines 167-191: The authors should move this paragraph to another new section as it is not about materials and methods but statistical analysis.

Author Response

We would like to thank the editor and the reviewers for their time and consideration. All comments and suggestions on the article helped us to make changes, which, we hope, will improve its content. Please find below a list of changes against each point which had been raised.

1. The title was changed according to the Reviewer’s suggestion.

2. We used four different questionnaires to measure four variables: early maladaptive schemas (YSQ-S3-PL), self-stigma (Self-Stigma Scale), psychological flexibility (AAQ-II), and self-compassion (Self-Compassion Scale Short). This matter was clarified in the text.

3. The section “2.1. Participants” was included under the Materials & Methods section. We now indicate the recruitment methods in the “Participants” section (ln 141-143): “Nonprobability sampling was used. The participants were recruited in Poland via the Internet on various forums and groups for patients and people with disabilities, using a snowball sampling.”

4. A new section “2.3. Statistical analysis” was added.